# Measuring and Modeling the Solubility of Hydrogen Sulfide in $r$FeCl$_3$/[bmim]Cl

**Huanong Cheng** [1],*  , **Na Li** [2], **Rui Zhang** [2], **Ning Wang** [2], **Yuanyuan Yang** [2], **Yun Teng** [2], **Wenting Jia** [2] and **Shiqing Zheng** [1]

[1] Institute of Computer and Chemical Engineering, Qingdao University of Science and Technology, Qingdao 266000, China; zheng_sq1@163.com

[2] College of Chemical Engineering, Qingdao University of Science and Technology, Qingdao 266000, China; li_na1@163.com (N.L.); zhang_r2@163.com (R.Z.); wang_n2@163.com (N.W.); yang_yy2@163.com (Y.Y.); teng_y1@163.com (Y.T.); Jia_wt1@163.com (W.J.)

* Correspondence: chn@qust.edu.cn

**Abstract:** The solubility of hydrogen sulfide in different mole ratios of ferric chloride and 1-butyl-3-methylimidazolium chloride ionic liquid ($r$FeCl$_3$/[bmim]Cl, $r$ = 0.6, 0.8, 1.0, 1.2, 1.4) at temperatures of 303.15 to 348.15 K and pressures of 100 to 1000 kPa was determined. The total solubility increased with the increase of pressure and the decrease of temperature. The solubility data were fitted using the reaction equilibrium thermodynamic model (RETM). The mean relative error between the predicted value and the measured value was less than 4%. Henry's coefficient and the equilibrium constant of chemical reaction at each temperature were calculated. Henry's coefficient first decreased and then increased with the increase of mole ratio, and increased with the increase of temperature. The equilibrium constant of the chemical reaction followed the same law as Henry's coefficient. The chemical solubility was related to both Henry's coefficient and the chemical equilibrium constant. H$_2$S had the highest chemical solubility in FeCl$_3$/[bmim]Cl at a mole ratio of 0.6 and a temperature of 333.15 K. The chemical solubility increased with the increase of pressure.

**Keywords:** ionic liquid; hydrogen sulfide; solubility; Henry's coefficient; reaction equilibrium constant





## 1. Introduction

Hydrogen sulfide (H$_2$S) is produced along with industrial production, natural gas development, biogas, and sludge treatment [1,2]. It is a toxic and harmful gas that not only causes damage to human health through odor and paralysis of nerves, but also damages the environment by forming acid rain. The liquid oxidation catalyst (LO-CAT) method [3] is a widely used wet oxidation desulfurization technology in which Fe(III) in the absorbent reacts with H$_2$S in the gas to become Fe(II), and H$_2$S becomes elemental sulfur. Then air is used to oxidize Fe(II) to generate Fe(III) to regenerate the absorbent. Compared with other desulfurization methods, the advantage of the LO-CAT method is that the raw material iron is cheap, easy to obtain, and harmless, and the desulfurization process is highly efficient [4]. However, the LO-CAT method still has some problems [5,6]: (1) Iron complex in the absorbent is easy to decompose, resulting in loss of oxidants. (2) The pH value of the absorbent solution has to be strictly controlled to prevent the iron ion from changing to iron hydroxide and iron sulfide precipitation during the reaction. (3) By-product salts, such as thiosulfate and sulfates, are formed due to overoxidation of H$_2$S in an alkaline or neutral solution. The by-product salts generated in the LO-CAT method process not only easily blocks the pipeline but also enters the wastewater, and it requires high energy consumption to treat the wastewater before it can be discharged into the environment.

In recent years, the green nature of ionic liquids has been widely recognized, and it is used as a solvent, catalyst, and extractant in various fields [7–10]. Due to the advantages of low vapor pressure and good solubility of ionic liquids, it is also used as a desulfurizer to

absorb $H_2S$. Some functional ionic liquids show excellent performance when absorbing $H_2S$ [4,11–15]. The iron-based ionic liquid (Fe-IL) 1-butyl-3-methylimidazolium tetrachloro-ferrate ([bmim]FeCl$_4$) is one of them [16]. The good solubility of ionic liquids ensures high solubility of $H_2S$. Fe(III) acts as an oxidant and a catalyst. Therefore, the triple functional ionic liquid has great potential in absorbing $H_2S$.

Yukihiro Yoshida and Gunzi Saito's research [17] on [bmim]FeCl$_4$ shows that its thermal decomposition temperature can reach 553 K. The high decomposition temperature ensures that ferric iron can stably exist in the form of complexes when oxidizing hydrogen sulfide. Wang Jianhong [18] reported that [bmim]FeCl$_4$ is both Lewis acidic and Bronsted acidic. $H_2S$ will not be overoxidized in an acidic environment [19]. Therefore, no thiosulfates and sulfates are produced during the oxidation of $H_2S$ by [bmim]FeCl$_4$, which avoids the production of salty wastewater. This was confirmed by detecting $H_2S$-loaded [bmim]FeCl$_4$ by infrared spectroscopy, in which no S–O bond absorption peaks were found [16]. The surface tension of [bmim]FeCl$_4$ is small, and the deposited sulfur is easy to be separated from the desulfurizer [18]. Although ionic liquids are more expensive, Wang's study shows that the use of [bmim]FeCl$_4$ to catalyze the oxidation of hydrogen sulfide is more economical than traditional LO-CAT desulfurization [20]. This is because the vapor pressure of the ionic liquid is negligible, the loss of desulfurization liquid is very small, and there is no side reaction resulting in the post-treatment cost. The above studies show that [Bmim]FeCl$_4$ has the advantages of thermal stability, acidity, and low operating cost. It is a potential desulfurizer for solving the shortcomings of the chelated iron method.

[bmim]FeCl$_4$ is synthesized by mixing anhydrous FeCl$_3$ and [bmim]Cl in an equimolar ratio. When the ratio is greater than 1, FeCl$_4^-$ anions react with excess FeCl$_3$ to form Fe$_2$Cl$_7^-$. When the ratio is less than 1, unreacted Cl$^-$ exists [21]. Therefore, there are different anion forms for different molar ratios. This paper uniformly uses $r$FeCl$_3$/[bmim]Cl to represent them, where $r$ represents the molar ratio of anhydrous FeCl$_3$ to [bmim]Cl. $r$FeCl$_3$/[bmim]Cl has similar properties to $r$AlCl$_3$/[bmim]Cl. Their properties can be changed by changing the concentration of metal ions (mole ratio $r$) [22]. Li Jiguang [23] reported the density, viscosity, and conductivity of FeCl$_3$/[bmim]Cl under different $r$'s at a temperature ranging from 293 to 363 K. Studies have shown that the viscosity decreases sharply as the iron ion concentration increases at the same temperature. The density increases linearly, and the conductivity first increases and then slowly decreases. In addition to density and viscosity, solubility is an indispensable physical property when studying the $H_2S$ absorption process. At present, the solubility data of hydrogen sulfide in $r$FeCl$_3$/[bmim]Cl are still lacking. The objectives of this work were to measure the solubility of hydrogen sulfide in $r$FeCl$_3$/[bmim]Cl ($r$ = 0.6, 0.8, 1.0, 1.2, 1.4) at temperatures from 303.15 to 348.15 K and pressures from 100 to 1000 kPa. A model was proposed to fit the experiment data. Henry's coefficient and the equilibrium constant of the chemical reaction were calculated. The effects of temperature, pressure, and iron content on the absorption were investigated.

## 2. Materials and Methods

### 2.1. Chemicals

$H_2S$ gas of 99% mass purity was supplied by the Yuyan Gas Company. Anhydrous FeCl$_3$ of 99% mass purity was obtained from the Aladdin Reagent Company. All of the above were used as purchased without further purification. [bmim]Cl of 97% mass purity was purchased from the Macklin Reagent Company and was further purified by recrystallization in acetonitrile–ethyl acetate [17,24]. After that, it was filtered before being dried in a vacuum dryer for 24 h, then placed in a desiccator. No obvious impurities in the sample were confirmed by H$^1$ NMR (Figure S1 in the Supporting Materials section) [25].

$r$FeCl$_3$/[bmim]Cl was synthesized by a well-established procedure [26]. A certain mass of [bmim]Cl was added in a three-necked flask with stirring and N$_2$ atmosphere, and heated to 60 °C. Then different molar ratios of anhydrous FeCl$_3$ were added. After stirring for 6 h, a dark brown liquid $r$FeCl$_3$/[bmim]Cl with a molar ratio of 0.6 to 1.4 was

obtained. When $r = 0.4$, the resultant ionic liquid is too viscous and not suitable for practical application. Therefore, mole ratio starts at $r = 0.6$ in this work. The ionic liquid was dried in a vacuum drying oven at 333 K for 48 h, and the water content was determined below the detection limit (100 ppm) of a Karl Fischer titrator (Mettler Toledo, DL31). The results (Figure S2 in the Supporting Materials section) of the infrared spectrum (Bruker Tensor 27) for the synthesized $r\text{FeCl}_3/[\text{bmim}]\text{Cl}$ were consistent with the literature [20,26], indicating that the structure of the synthesized iron-based ionic liquid cation was not changed. There was no –OH stretching vibration peak of 3300 cm$^{-1}$ in the spectrum, indicating that the ionic liquid did not contain water. The ionic liquid was put into a desiccator with silica gel for later use.

### 2.2. Experimental Design

The experimental apparatus used to measure the solubility of $H_2S$ in $r\text{FeCl}_3/[\text{bmim}]\text{Cl}$ was homemade and is shown in Figure 1. It consisted of a vacuum pump (1 in Figure 1, the same below), a high-pressure nitrogen cylinder (12), a high-pressure hydrogen sulfide cylinder (11), a buffer tank (13), a phase equilibrium kettle (16) made of Hastelloy (SL-Q100, Sen Long) with a magnetic-drive stirrer, and a tail gas absorption device (17). The volumes of the buffer tank and phase equilibrium kettle were 510.7 mL ($V_1$) and 122.8 mL ($V_2$), respectively. They were placed in two constant-temperature circulating water baths (14, 15) (HH-Sc, Changzhou Langyue Instrument Manufacturing Co., Ltd., Hangzhou, Zhejiang Province, China), respectively, and the temperature inside the kettle was measured by a TES 1320 TYPE-K thermocouple (9, 10) with an uncertainty of 0.03 K. The pressures in the buffer tank and the phase equilibrium kettle were measured by pressure sensors (2, 3) (YK-100B, −100–5000 kPa, PR Software V1.5, Xi'an Yunyi Instrument Co., Ltd., Xi 'an, Shaanxi Province, China) with an uncertainty of 3 kPa and transmitted to the computer.

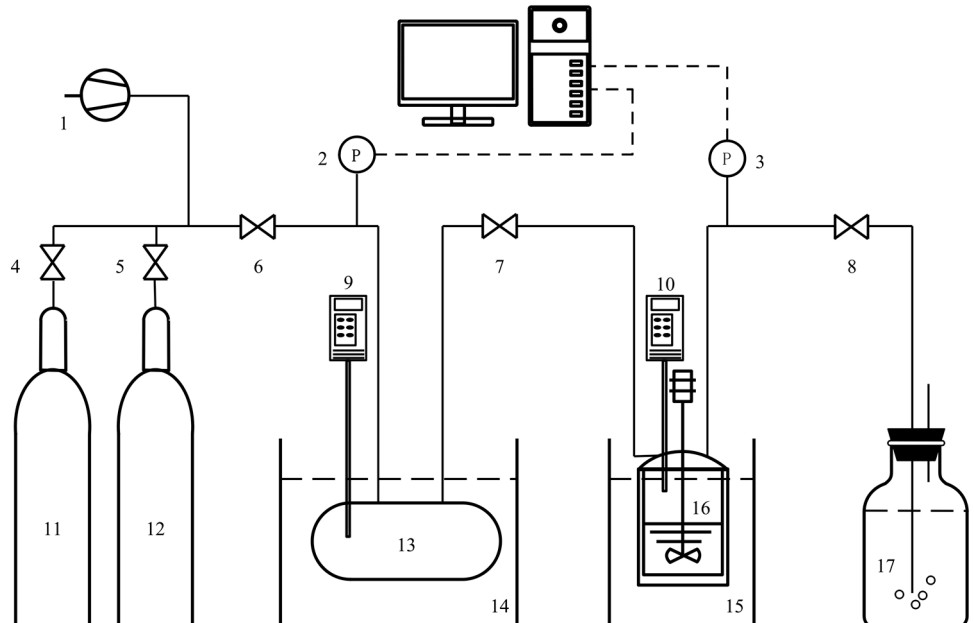

**Figure 1.** Scheme of the experimental setup: 1, vacuum pump; 2, 3, pressure sensor; 4 to 8, valve; 9, 10, thermocouple; 11, H$_2$S gas cylinder; 12, N$_2$ gas cylinder; 13, buffer tank; 14, 15, water bath; 16, equilibrium kettle; 17, gas scrubber.

The operation of the apparatus was started by introducing a known mass of a Fe-IL weighed with an electronic balance (TG328-A, Shanghai Balance Instrument Factory) with an uncertainty of $0.3 \times 10^{-7}$ kg to the phase equilibrium kettle, and the whole equipment (except the gas cylinder) was evacuated to pressures below 1.0 kPa by the vacuum pump, and the temperature was raised to 348 K. The Fe-IL inside the kettle was kept at this

temperature under vacuum for at least 4 h to remove trace amounts of water and volatile impurities. The temperature was then adjusted to the required value through the water bath, and a known amount of $H_2S$ was introduced to the buffer tank from the $H_2S$ gas cylinder. After 30 min of heat preservation, the pressure ($P_1$) in the buffer tank was recorded. Next, the gas from the buffer tank was released into the phase equilibrium kettle, and the remaining pressure in the buffer tank ($P_2$) was recorded after the valve was closed. Then the stir was turned on. The sign of reaching phase equilibrium was that the pressure in the equilibrium kettle remained unchanged for 1 h, and equilibrium pressure ($P_3$) was recorded. After the experiment, $H_2S$ in the system was discharged into a saturated sodium hydroxide solution and purged with nitrogen for a long time.

The total solubility of $H_2S$ in the Fe-IL solvent based on molality, $m_{H_2S,t}$, can be determined from the following equation:

$$m_{H_2S,t} = \frac{n_t}{w_{Fe-IL}} \tag{1}$$

where $w_{Fe-IL}$ is the mass of $r\text{FeCl}_3/[\text{bmim}]\text{Cl}$ added to the phase-balanced kettle; $n_t$ denotes the integral molar quantity of $H_2S$, which is absorbed into the liquid at phase equilibrium; and $n_t$ can be calculated by the difference between the molar quantity of gas filled into the phase equilibrium kettle and the molar quantity remaining at the equilibrium gas phase:

$$n_t = n_{add} - \frac{P_{H2S} \cdot \left(V_2 - \frac{w_{Fe-IL}}{\rho_{Fe-IL}(T)}\right)}{Z_3 \cdot R \cdot T} \tag{2}$$

where $P_{H2S}$ is the equilibrium partial pressure of $H_2S$ in the equilibrium kettle, kPa; $\rho_{Fe-IL}(T)$ denotes the density of $r\text{FeCl}_3/[\text{bmim}]\text{Cl}$ liquid in the phase equilibrium kettle at temperature $T$; $\rho_{Fe-IL}(T)$ is obtained according to the literature [23] (density data are shown in Table S1 in the Supporting Materials section); and $R$ is gas constant with a value of $8.314 \, \text{J} \cdot \text{mol}^{-1} \cdot \text{K}^{-1}$.

$P_{H2S}$ can be calculated by

$$P_{H2S} = P_3 - P_0 - P_L, \tag{3}$$

where $P_0$ is the pressure after the system is evacuated, and $P_L$ is the partial pressure of the solvent in the equilibrium kettle.

For the Fe-IL in this work, the vapor pressure was very low, so $P_L = 0$. For the water solvent in the verification experiment, $P_L$ cannot be ignored.

The molar quantity of $H_2S$ filled into the phase equilibrium kettle, $n_{add}$, can be obtained by the real gas state equation:

$$n_{add} = \frac{\left(\frac{P_1}{Z_1} - \frac{P_2}{Z_2}\right) \cdot V_1}{R \cdot T}, \tag{4}$$

where $Z_1$, $Z_2$, and $Z_3$ are compression factors of $H_2S$ at temperature $T$ and pressures $P_1$, $P_2$, and $P_3$, respectively. These compression factors are calculated using the Soave–Redlich–Kwong equation in Aspen Plus software (aspenONE® for Universities, Aspen Technology, Inc., Bedford, MA, USA).

## 3. Reaction Equilibrium Thermodynamic Model

Henry's coefficient is one of the important thermodynamic coefficients, which reflects the physical dissolution degree of the gas solute in the solvent. Henry's law [12,27] or the Krichevsky–Kasarnovsky equation [28] is usually used directly to estimate Henry's coefficient. The solubility used in these equations is physical. However, the solubility measured in this paper includes both physical dissolution and chemical dissolution, so the above equations cannot be used directly to estimate Henry's coefficient. Huang Kuan [29] put forward the reaction equilibrium thermodynamic model (RETM) for the estimation of

Henry's coefficient with both physical absorption and chemical absorption. This model introduces a reaction equilibrium constant to strip chemical dissolution. In this work, a new RETM model based on the absorption and reaction of $H_2S$ in $rFeCl_3$/[bmim]Cl (Fe (III)-IL) was derived as follows: $H_2S$ is first absorbed into the ionic liquid. Since Fe (III)-IL is acidic, $H_2S$ is not decomposed into $HS^-$ or $S^{2-}$ ionic state, but exists as an $H_2S$ molecular state in liquid [30]. The $H_2S$ molecule in the liquid undergoes redox reaction with Fe (III)-IL. It was reported [20] that when measuring the sulfur capacity, a large amount of iron is not reduced, so the redox reaction can be regarded as a reversible reaction. According to the literature [16,23], the reaction equation of $H_2S$ being absorbed and oxidized by Fe (III)-IL is as follows:

$$H_2S_{(g)} \rightleftharpoons H_2S_{(l)}, \tag{5}$$

$$H_2S_{(l)} + 2Fe(III) - IL_{(l)} \rightleftharpoons S_{(s)} + 2HFe(II) - IL_{(l)}, \tag{6}$$

where $g$, $l$, and $s$ are gas, liquid, and solid phases, respectively. Equation (5) means that $H_2S$ in the gas phase is captured by the ionic liquid and then dissolved in the liquid phase, which obeys Henry's law:

$$P_{H2S} = H_m \cdot \gamma_{H_2S} \cdot \frac{m_{H_2S}}{m^0}, \tag{7}$$

where $P_{H2S}$ is the partial pressure of $H_2S$. The vapor pressure of the ionic liquid is negligible. Therefore, the gas phase in the phase equilibrium kettle is pure $H_2S$ without solvent vapor. $H_m$ denotes Henry's coefficient based on molality. $m_{H_2S}$ denotes the molality of the physically dissolved $H_2S$ in liquid phase. $m^0$ and $P^0$ are standard molality with a value of 1 mol/kg and standard pressure with a value of 100 kPa, respectively. $\gamma_{H_2S}$ is the activity coefficient of $H_2S$ physically dissolved in the liquid phase. Equation (6) reflects the process in which $H_2S$ in the liquid phase is oxidized to sulfur by Fe(III), while Fe(III) is reduced to Fe(II). The chemical reaction equilibrium constant $K$ of this reaction is expressed as follows:

$$K = \frac{\left(\frac{m_{Fe(II)}}{m^0}\right)^2}{\left(\frac{m_{Fe(III)}}{m^0}\right)^2 \cdot \frac{m_{H_2S}}{m^0}}, \tag{8}$$

where $m_{Fe(II)}$ and $m_{Fe(III)}$ denote the molality of Fe(II)-IL and Fe(III)-IL, respectively. The mass balance equations for $H_2S$ and ionic liquids are given as follows:

$$m_{H_2S,t} = m_{H_2S} + m_{H_2S,r}, \tag{9}$$

$$m_{Fe-IL} = m_{Fe(III)} + m_{Fe(II)}, \tag{10}$$

where $m_{H_2S,r}$ is the molality of $H_2S$ involved in the reaction. According to the stoichiometric ratio, $m_{H_2S,r} = \frac{1}{2}m_{Fe(II)}$. $m_{Fe-IL}$ is the initial molality of the ionic liquid, which is represented by Equation (11).

$$m_{Fe-IL} = \frac{1}{M_{Fe-IL}}, \tag{11}$$

where $M_{Fe-IL}$ is the molar mass of ionic liquid.

$rFeCl_3$/[bmim]Cl was actually a mixture except for $r = 1$. The average molar mass needed to be calculated. The calculation method is listed in Table 1.

**Table 1.** Calculation method for the molar mass ($M$) of different composition ratios of $rFeCl_3$/[bmim]Cl.

| | Composition | Computational Formula |
|---|---|---|
| $r < 1$ | [bmim]Cl, [bmim]FeCl$_4$ | $M_{Fe-IL} = r \cdot M_{[bmim]FeCl_4} + (1-r) \cdot M_{[bmim]Cl}$ |
| $r = 1$ | [bmim]FeCl$_4$ | $M_{Fe-IL} = M_{[bmim]FeCl_4}$ |
| $r > 1$ | [bmim]FeCl$_4$, [bmim]Fe$_2$Cl$_7$ | $M_{Fe-IL} =$ $(2-r) \cdot M_{[bmim]FeCl_4} + (r-1) \cdot M_{[bmim]Fe_2Cl_7}$ |

Assuming that the liquid phase is an ideal solution, the value of $\gamma_{H_2S}$ is set to 1. Combine Equations (7) to (11) together, and sort out the new RETM model:

$$m_{H_2S,t} = \frac{P_{H2S}}{H_m} + \frac{\sqrt{H_m \cdot P_{H2S} \cdot K} - P_{H2S} \cdot K}{2H_m - 2P_{H2S} \cdot K} m_{Fe-IL} \tag{12}$$

According to Equation (12), take $P$ as the independent variable and $m_{H_2S,t}$ as the dependent variable, and the values of unknown quantity $H_m$ and $K$ can be obtained by using nonlinear regression to fit the solubility data in ORIGIN software. The accuracy of the model was characterized by mean relative error (*MRE*) and maximum relative error (*MD*).

$$MRE = \frac{100}{N} \sum_{i=1}^{N} \left| \frac{m_i^{\exp} - m_i^{fit}}{m_i^{\exp}} \right|, \tag{13}$$

$$MD = Max \left( 100 \left| \frac{m_i^{\exp} - m_i^{fit}}{m_i^{\exp}} \right| \right), \tag{14}$$

where $N$ is the number of experiments per temperature, $m_i^{\exp}$ is the total solubility measured in the experiment, and $m_i^{fit}$ is the total solubility calculated using the parameters obtained from the RETM model.

## 4. Results and Discussion

### 4.1. Experimental Method Validation

The solubility of $H_2S$ in water was measured at temperatures 308.15, 323.15, and 343.15 K to verify the reliability of the experimental method. For the convenience of comparison with literature data, the solubility of $H_2S$ in water is calculated by Equation (15) based on the molar fraction. The data obtained from the experiment are presented in the form of points in Figure 2. Henry's coefficients of $H_2S$ in water [31] are reported to be 68,495.7 kPa at 308.15 K, 89,571.3 kPa at 323.15 K, and 120,576.8 kPa at 343.15 K. As shown in Figure 2, the lines with the slope of Henry's coefficients are drawn in a coordinate system with the equilibrium pressure of $H_2S$ on the horizontal axis and the solubility on the vertical axis. The relative percentage deviations between the experimental values and the literature values at the three temperatures are all less than 3%, so the device has the ability to measure the solubility.

$$x_{H_2S} = \frac{n_t \cdot M_{H_2O}}{w_{H_2O}} \tag{15}$$

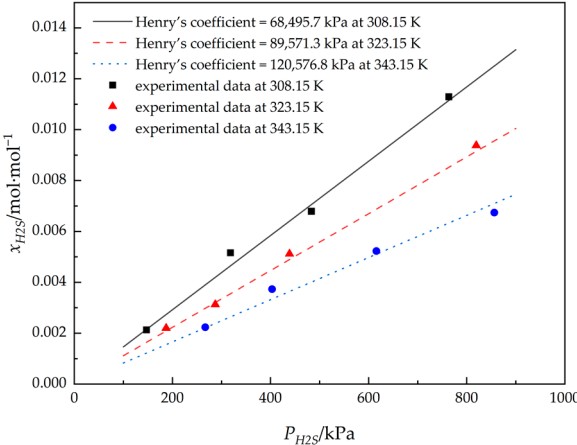

**Figure 2.** Comparison of experimental values and literature values of $H_2S$ solubility in water.

### 4.2. H₂S Solubility in rFeCl₃/[bmim]Cl

The solubility of $H_2S$ in $r$FeCl$_3$/[bmim]Cl ($r$ = 0.6, 0.8, 1.0, 1.2, 1.4) at temperatures of 303.15, 318.15, 333.15, and 348.15 K and pressures up to about 1.0 MPa is summarized in Figure 3. For detailed solubility data with uncertainty, see Table S2 in the Supporting Materials section. As can be seen, the total solubility of $H_2S$ increases with increasing pressure and decreasing temperature at the same $r$, which is similar to the solubility of most gases. It not only shows that our experimental device is reliable, but also indicates that the phase equilibrium of iron-based ionic liquid and $H_2S$ follows the same law as other traditional solvents.

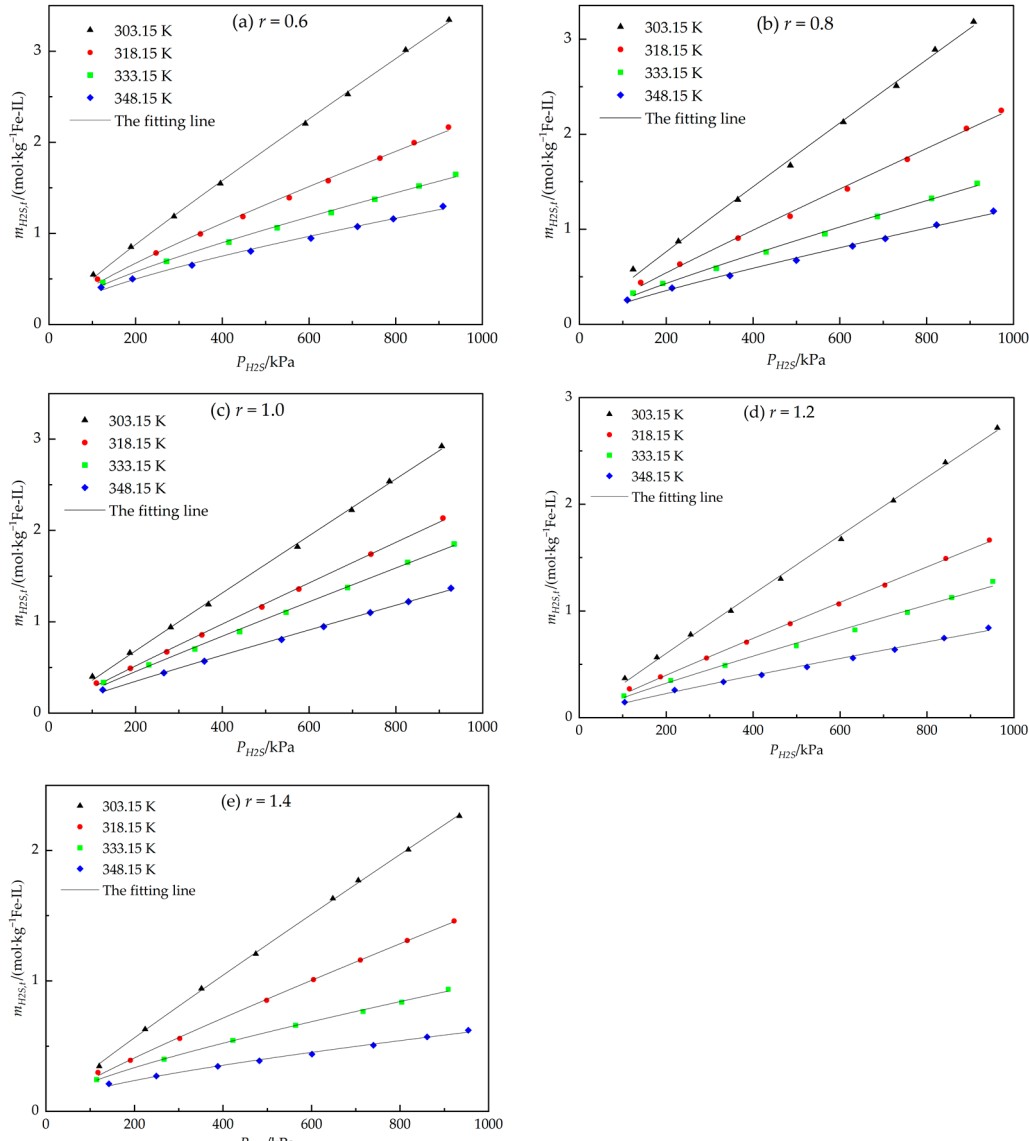

**Figure 3.** Experimental results for the solubility of H₂S in $r$FeCl₃/[bmim]Cl: (**a**)$r$=0.6; (**b**)$r$=0.8; (**c**)$r$=1.0; (**d**)$r$=1.2; (**e**)$r$=1.4.

### 4.3. Modeling

The RETM model was used to fit the data of total solubility and pressure, and the fitting curve is shown in Figure 3. Henry's coefficients and the equilibrium constants calculated by the RETM model are listed in Table 2. The *MRE* of each group of forecasts does not exceed 4%, which indicates that the data predicted by the RETM model are in good agreement with the experimental data. The *MD* is 9.77%, which indicates that the requirement of engineering calculation is met.

**Table 2.** Henry's coefficients ($H_m$) and equilibrium constants ($K$) obtained by RETM model fitting and the mean relative error (*MRE*) and maximum relative error (*MD*) between the experiments and the fitting values at temperature $T$ [a].

| | *T*/K | | | |
|---|---|---|---|---|
| | 303.15 | 318.15 | 333.15 | 348.15 |
| *r* = 0.6 | | | | |
| $H_m$/kPa | 325.7 | 604.9 | 966.6 | 1408.7 |
| *K* | 0.04588 | 0.1585 | 0.3091 | 0.4082 |
| *MRE*/% | 1.66 | 2.08 | 1.80 | 2.61 |
| *MD*/% | 6.70 | 8.73 | 5.65 | 8.20 |
| *r* = 0.8 | | | | |
| $H_m$/kPa | 313.6 | 505.7 | 832.7 | 1174.4 |
| *K* | 0.0108 | 0.0242 | 0.0713 | 0.0940 |
| *MRE*/% | 2.49 | 2.74 | 2.80 | 3.01 |
| *MD*/% | 7.77 | 8.09 | 7.95 | 7.75 |
| *r* = 1.0 | | | | |
| $H_m$/kPa | 330.8 | 471.6 | 575.9 | 803.4 |
| *K* | 0.0049 | 0.0103 | 0.0175 | 0.0205 |
| *MRE*/% | 3.17 | 1.95 | 2.56 | 1.78 |
| *MD*/% | 9.77 | 7.56 | 8.48 | 7.83 |
| *r* = 1.2 | | | | |
| $H_m$/kPa | 378.8 | 638.1 | 935.8 | 1484.8 |
| *K* | 0.0063 | 0.0141 | 0.0359 | 0.0385 |
| *MRE*/% | 2.84 | 1.86 | 3.11 | 3.07 |
| *MD*/% | 9.35 | 9.05 | 6.67 | 6.05 |
| *r* = 1.4 | | | | |
| $H_m$/kPa | 462.0 | 797.5 | 1697.3 | 3492.7 |
| *K* | 0.0322 | 0.0855 | 0.3794 | 0.4917 |
| *MRE*/% | 1.39 | 1.62 | 0.96 | 2.34 |
| *MD*/% | 5.73 | 8.05 | 1.81 | 7.63 |

[a] $T$ is the equilibrium temperature in the equilibrium kettle, K. $r$ is the molar ratio of anhydrous $FeCl_3$ to [bmim]Cl. $H_m$ is Henry's coefficient of $H_2S$ in Fe-IL based on molality, kPa. $K$ is the chemical reaction equilibrium constant. *MRE* is mean relative error. *MD* is maximum relative error.

### 4.4. Influence of Molar Ratio

Henry's coefficients ($H_m$) at different temperatures and the molar ratio ($r$) obtained from Table 2 are plotted. The variation of $H_m$ with $r$ at different temperatures is shown in Figure 4. As we can see, with the molar ratio $r$ = 1.0 of $FeCl_3$ and [bmim]Cl as the limit, $H_m$ shows different trends for the composition of $r \leq 1.0$ and $r \geq 1.0$. $H_m$ decreases with the increase of $r$ when $r \leq 1.0$. Li Jiguang [23] measured the viscosity data of $rFeCl_3$/[bmim]Cl at 298.15 K. The viscosity decreased from close to 200 to 31.2 mPa·s when $r$ increased from 0.6 to 1.0. Due to the sharp drop in viscosity, $H_2S$ diffuses more freely in the liquid, and $H_2S$ also moves more easily from the gas phase into the liquid phase [32]. Therefore, the physical solubility increases with the increase of iron content when $r \leq 1.0$.

On the contrary, $H_m$ increases with the increase of $r$ when $r \geq 1.0$. The data measured by Li Jiguang [23] shows that the viscosity decreased from 31.2 to 27.4 mPa·s when $r$ increased from 1.0 to 1.5 at 298.15 K. The viscosity of $rFeCl_3$/[bmim]Cl changes very little, which has little effect on the absorption of $H_2S$. While the density of the liquid continues to increase, from about 1.32 to about 1.42 g·cm$^{-3}$, the free volume of the ionic liquid decreases to hold fewer $H_2S$ molecules [33]. Therefore, the physical solubility decreases with the increase of iron content when $r \geq 1.0$.

The chemical reaction equilibrium constant ($K$) at different temperatures and the molar ratio ($r$) obtained from Table 2 are plotted. The variation of $K$ with $r$ at different temperatures is shown in Figure 5. Likewise, $K$ shows a similar law to $H_m$. As shown in Figure 5, $K$ decreases with the increase of $r$ when $r \leq 1.0$. In the ionic liquids consisting of $r \leq 1.0$, the types of anions are $Cl^-$ and $FeCl_4^-$. $Cl^-$ is a neutral ion, whereas $FeCl_4^-$ is an acidic ion. Therefore, as $r$ increases, the concentration of acid anion increases, which

is not conducive to the combination with acid molecule $H_2S$, resulting in a decrease in chemical reaction.

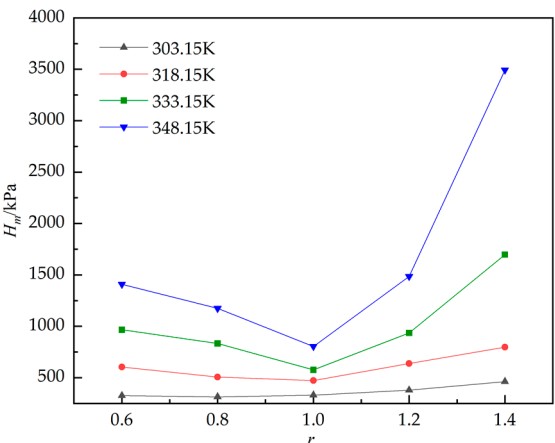

**Figure 4.** The effect of *r* on Henry's coefficient at different temperatures.

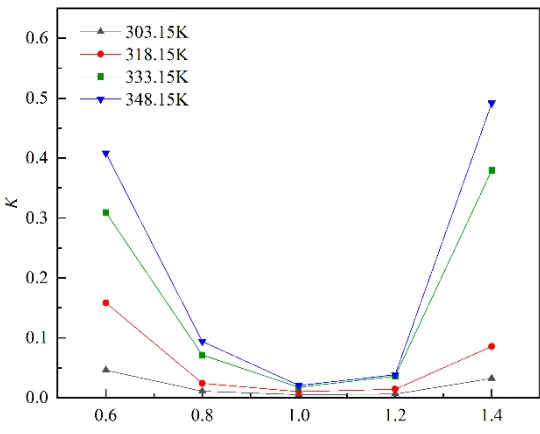

**Figure 5.** The effect of *r* on the chemical equilibrium constant at different temperatures.

In the case of $r \geq 1.0$, *K* increases with the increase of *r*. In ionic liquids composed of $r \geq 1.0$, the types of anions are $FeCl_4^-$ and $Fe_2Cl_7^-$. Both of them are acidic anions [24]. Nguyen et al. [34] reported that $Fe_2Cl_7^-$ exhibited higher catalytic activity than $FeCl_4^-$. Therefore, the chemical reaction is enhanced.

Chemical solubility is the amount of $H_2S$ dissolved in the absorber for a chemical reaction. Consequently, the chemical solubility is determined by Henry's coefficient $H_m$ and the equilibrium constant *K*. The oxidation of hydrogen sulfide to sulfur by $rFeCl_3/[bmim]Cl$ is a reversible reaction. The amount of products is related to the concentration of reactants and the degree of reaction. The greater the concentration of the reactant and the degree of the reaction, the greater the amount of the product. $H_m$ and *K* change with the iron content in the same law; that is, the concentration of reactants first increases and then decreases with the content of iron, while the reaction degree first decreases and then increases with the content of iron. This indicates that it is an offsetting process. The physical solubility can be calculated by Henry's coefficient from RETM, and the chemical solubility can be obtained by subtracting the physical solubility from the total solubility. In the experiment, it is difficult to measure solubility at the same equilibrium pressure at each temperature and iron content. For the convenience of comparison, the chemical solubility at 200 kPa is calculated by RETM. The changes in chemical solubility with *r* at different temperatures are graphically presented in Figure 6. The chemical solubility decreases first and then increases with the increase of *r*. The maximum chemical solubility appears when *r* = 0.6. For smaller

molar ratios *r*, it is not suitable for application due to very high viscosity [32] in the test temperature range.

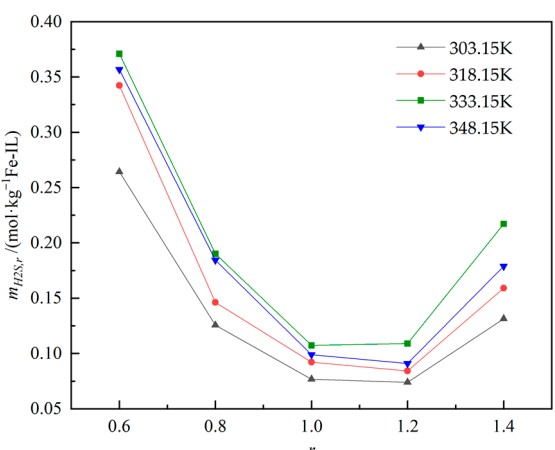

**Figure 6.** Influence of iron content on chemical solubility at different temperatures at 200 kPa.

After the $r$FeCl$_3$/[bmim]Cl ionic liquid absorbs H$_2$S, the regeneration of the absorbent (ionic liquid) is the oxidation of Fe(II) to Fe(III) by oxygen, instead of desorption by rectification. The chemical solubility is the factor that measures the capacity of H$_2$S absorption. Therefore, $r$ = 0.6 is the best molar ratio of anhydrous FeCl$_3$ to [bmim]Cl.

*4.5. Influence of Pressure*

Physical solubility and chemical solubility increase with the increase of pressure. For a given temperature and molar ratio, Henry's coefficient $H_m$ and equilibrium constant $K$ are fixed values. Physical solubility is directly proportional to pressure. The higher the pressure, the more dissolved H$_2$S, the more converted H$_2$S, and the greater the chemical solubility. For example, Figure 7 illustrates the physical solubility and chemical solubility of FeCl$_3$/[bmim]Cl at a mole ratio of 0.6 and at 333.15 K. Both physical and chemical solubilities increase as the pressure increases. Physical solubility changes more significantly. Under low pressure, the physical solubility is lower than the chemical solubility. Additionally, with the increase of pressure, the physical solubility gradually exceeds the chemical solubility.

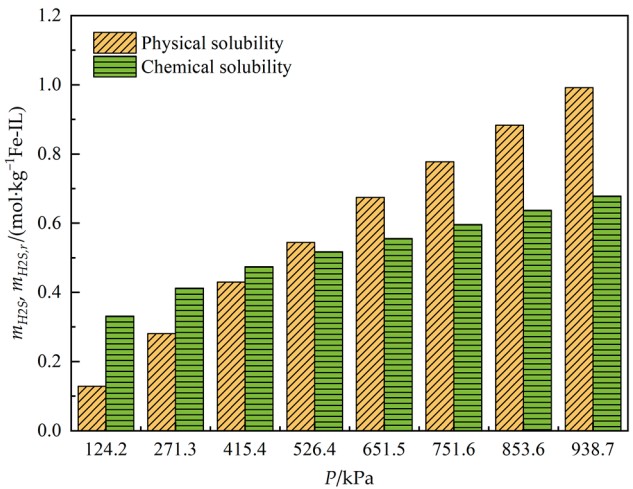

**Figure 7.** Physical and chemical solubilities of H$_2$S in FeCl$_3$/[bmim]Cl at a mole ratio of 0.6 and at 333.15 K.

### 4.6. Influence of Temperature

Henry's coefficient $H_m$ increases as the temperature rises under the same molar ratio and pressure. It means that the physical solubility decreases as the temperature increases. A rise in temperature intensifies molecular motion, making the molecules harder to fix in the liquid phase and thus not conducive to physical dissolution. The equilibrium constant of chemical reaction, *K,* increases with the increase of temperature. The increase of *K* indicates that the temperature rise causes the reaction to move in the positive direction. This means that the reaction of generating elemental sulfur is endothermic. However, the increase of *K* does not mean a greater chemical solubility, because only dissolved $H_2S$ participates in the chemical equilibrium reaction. Although the *K* value is high, most of the dissolved $H_2S$ is converted to elemental sulfur. However, the amount of $H_2S$ converted is still small when the amount of $H_2S$ dissolved is small. As a consequence, this leads to a low chemical solubility. For instance, the physical and chemical solubilities of $H_2S$ in $FeCl_3$/[bmim]Cl at a mole ratio of 0.6 and at 200 kPa with temperature change are shown in Figure 8. With the increase of temperature, the physical solubility decreased from 0.61 to 0.14 mol·kg$^{-1}$ Fe-IL, and the chemical solubility first increased from 0.26 to 0.37 mol·kg$^{-1}$ Fe-IL and then decreased to 0.35 mol·kg$^{-1}$ Fe-IL. The chemical solubility reaches its maximum at 333.15 K. This explains that the chemical solubility is not only related to the equilibrium constant but also influenced by Henry's coefficient.

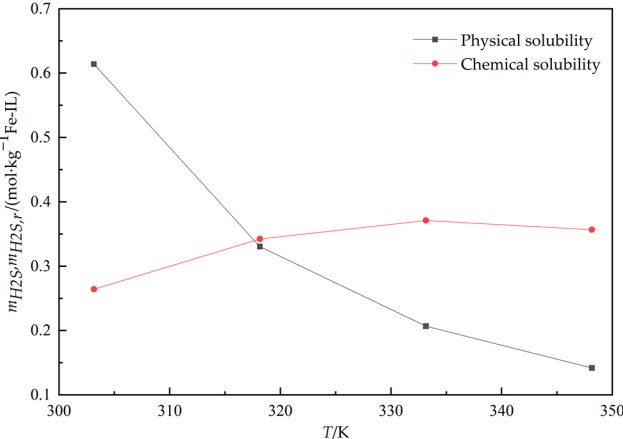

**Figure 8.** Effects of temperature on physical and chemical solubilities in $FeCl_3$/[bmim]Cl at a mole ratio of 0.6 and at 200 kPa.

### 5. Conclusions

The experimental data for the solubility of $H_2S$ in $rFeCl_3$/[bmim]Cl (*r* = 0.6, 0.8, 1.0, 1.2, 1.4) were measured and presented in this work at temperatures from 303.15 to 348.15 K and pressures from 100 to 1000 kPa. The solubility of $H_2S$ in $rFeCl_3$/[bmim]Cl was well predicted by using the reaction equilibrium thermodynamic model. The chemical solubility of $H_2S$ at $rFeCl_3$/[Bmim]Cl is highest at 333.15 K. Additionally, the higher the pressure is, the greater the chemical solubility will be. These conclusions are helpful for $H_2S$ absorption process simulation and process design in $rFeCl_3$/[bmim]Cl.

**Supplementary Materials:** The following are available online at https://www.mdpi.com/article/10.3390/pr9040652/s1: Figure S1: NMR of [bmim]Cl, Figure S2: Infrared spectrum of $rFeCl_3$[bmim]Cl, Table S1: Density data from the literature, Table S2: Total solubility ($m_{H2S,t}$) of $H_2S$ in different composition ratios of rFeCl$_3$/[bmim]Cl.

**Author Contributions:** Conceptualization, H.C. and N.L.; data curation, N.L., R.Z., N.W., Y.Y., and W.J.; formal analysis, H.C., N.L., and R.Z.; funding acquisition, H.C.; methodology, H.C. and N.L.; project administration, H.C.; resources, H.C. and S.Z.; supervision, H.C. and S.Z.; validation, N.L., R.Z., N.W., Y.Y., Y.T., and W.J.; visualization, N.L., R.Z., N.W., Y.Y., and Y.T.; writing—original draft, H.C. and N.L. All authors have read and agreed to the published version of the manuscript.

**Funding:** National Natural Science Foundation of China: 21775081, Natural Science Foundation of Shandong Province: ZR2020MB145, Key Technology Research and Development Program of Shandong: 2018GGX107004.

**Institutional Review Board Statement:** Not applicable.

**Data Availability Statement:** The data presented in this study are available on request from the corresponding author.

**Conflicts of Interest:** The authors declare no conflict of interest.

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
