# Peer review of "Measuring and Modeling the Solubility of Hydrogen Sulfide in rFeCl3/[bmim]Cl"

_processes, doi:10.3390/pr9040652_

Round 1

Reviewer 1 Report

Recommendation: Publish after major revisions noted.

Comments:

Authors have measured at different temperatures and pressures, the solubility of hydrogen sulfide in different mole ratio of ferric chloride and 1-Butyl-3-methylimidazolium chloride ionic. The solubility data were fitted using the Reaction Equilibrium Thermodynamic model. This paper contains interesting experimental data. However, I have some notes:

1)      In section 2.1, the purity of the compounds (H2S, Anhydrous FeCl3 and [bmim]Cl)  are not specified (mass fraction or mole fraction?).

2)      In Figure 2, the y axis represents the solubility of H2S in water, however the paper treats the solubility as molality. In this Figure the unities of the solubility is not well defined, also the experimental and the literature data are not mentioned which are, lines or dots? The legend of Figure 2 should be modified, there are information in theFigure that are missing in the legend.

3)      In Table 2, the mH2S,t ± uncertainties unities are missing For example it could be: mH2S,t ± u(H2S,t)/mol H2S kg-1IL

Author Response

Thank you very much for the comments. We have made changes to the article. The changes are shaded in yellow in the manuscript. Here are our answers to your questions.

1)      In section 2.1, the purity of the compounds (H2S, Anhydrous FeCl3 and [bmim]Cl)  are not specified (mass fraction or mole fraction?).

They are mass fraction. We have noted this in the revised manuscript.

2)      In Figure 2, the y axis represents the solubility of H2S in water, however the paper treats the solubility as molality. In this Figure the unities of the solubility is not well defined, also the experimental and the literature data are not mentioned which are, lines or dots? The legend of Figure 2 should be modified, there are information in the Figure that are missing in the legend.

We have added Equation (15) to the manuscript to define the solubility based on mole fraction.

We also added the legend in Figure 2 and the corresponding text in Section 4.1.

3)      In Table 2, the mH2S,t ± uncertainties unities are missing for example it could be: mH2S,t ± u(H2S,t)/mol H2S kg-1IL

We have added units of uncertainty to Table 2. In addition, according to the opinion of another reviewer, we moved Table 2 to the supporting materials to make the manuscript more concise.

Reviewer 2 Report

The work that is presented in the manuscript entitled “Measuring and modeling the solubility of hydrogen sulfide in rFeCl3/[bmim]Cl” by Cheng et al. is an overall interesting work, but some improvement should be undertaken.

In this work the authors measured the solubility of H2S in the ionic liquid rFeCl3/[bmim]Cl at different molar ratios, temperatures and pressures. Furthermore, they modelled the experimental data with the Reaction Equilibrium Thermodynamic Model. The findings are interesting and fit well with the already published literature.

However, in my opinion some general and specific points should be reconsidered the authors (mainly about the novelty of the work, text flow and English language, and better presentation of the results), before accepting the present work for publication. My comments on those general and specific points are listed below.

General comments:

- The novelty of the work is not very clear. The novelty of the work should be clarified in the Introduction section.

- The text flow and the presentation of the results should be improved (see specific comments). Moreover, in some parts of the text more discussion should be added to clarify what the authors try to convey.

- For all tablets and figures leave appropriate space between the text and ensure that the legend and corresponding figure or table appear at the same page.

- The English language needs improvement, several mistakes e-g. misuse of past and present tense, and of passive and active voice can be read in the text.

- Please add the “Contribution of authors” and “Conflict of interests” sections.

Specific comments:

- line 5: Please add full affiliation. Also include emails of all authors.

- line 16: Replace “has” by “follows”

Introduction

- line 34: Please correct reference [4] from subscript form.

- lines 34-36: Please reform that sentence, it is not very clear.

- line 38: Please correct “salt” to “salts”

- lines 56-57: Please reform that sentence, it is not very clear.

- line 58: Correct “was” to “were”.

- line 59: Please delete “also”.

- line 69: Please place a comma before “where”.

- line 80: Correct “was” to “is”.

Materials and Methods

- lines 86-88: Please replace “with 0.99 purity” by “of 99% purity” or “with 0.97 purity” by “of 97% purity”.

- line 91: What was the RH% conditions of the desiccator? Was a saturated salt solution used?

- line 92: Please indicate that the H1 NMR spectra can be found at the supplementary materials.

- line 101: Please indicate that the IR spectra can be found at the supplementary materials.

- line 106: The subsection title could be improved.

- line 107: The apparatus described here, is a custom-made in-house equipment? If so, it should be mentioned.

- lines 108-111: Please indicate those parts described here in Fig. 1.

- lines 111-112: This sentence is missing a verb.

- line 118: Which software was used for the acquisition of the data?

- lines 134-138: Please reform those sentences, avoiding beginning the sentence with “When”.

- line 151: Please add the equation in the text or describe the calculation procedure.

- line 164: What was the version of the Aspen Plus software? Who is the supplier?

- line 218: Does ADD stand for mean relative error? If so, please clarify it by using more appropriate abbreviation.

Results and Discussion

- line 229: Please delete “reference” before “[31]”.

- line 231: Please replace “put” by “presented”.

- line 232: Are those literature values presented? What do you mean with this point?

- lines 240-242: Please further discuss this point. Does this point also indicates/verifies the reliability of the experimental method that was followed in the present work?

- line 243, Table 2: Is this Table really helpful? Does it give significant information rather than Fig. 3. If yes, please discuss the Table in the text. If no, I would suggest removing the Table. Please note that if you decide to remove Table 2, I would suggest that the statistical presentation of the results should be transferred to Fig. 3.  

- line 247, Figure 3: Please improve the legend of the figure by indicating the fitting lines also presented. Moreover, please move the title of each subfigure above of each subfigure.

- line 251: Please correct “was” to “is”.

- lines 251-253: Please discuss this point. What the values of ADD and MD suggest?

- line 262: Please introduce Fig. 4 in the text, before referring to it.

- line 279: Please introduce Fig. 5 in the text, before referring to it.

- line 285: Please capitalize the first letter of the sentence. I would suggest reformation of the sentence and avoiding beginning with “When”.

- line 306: Please replace “showed” by “presented”.

- line 308: The viscosity values were measured or taken by the literature? If so, please add reference.

- line 325: Please capitalize the first letter of the subsection title.

- line 339: Was r=0.6 was chosen on purpose for this instance?

Conclusions

- line 355: Please correct “are analyzed” to “were analyzed”.

- line 360: Please delete the “As can be seen from Figure 6 and Figure 8” part. Conclusions should highlight the most important findings of the work and should not refer to previous parts of the work.

Reviewer 3 Report

This study reported the solubility of hydrogen sulfide in rFeCl3/[bmim]Cl.  However, this work shows a low novelty in the current version. I do not think that this paper can be published in Processes.

 1. The article is not innovative. Compared with the latest similar research in terms of innovation, what progress has been made in this article? There are several papers focusing on H2S solubility measurements and predictions in ILs published in the literature (Estimating hydrogen sulfide solubility in ionic liquids using a machine learning approach; Evaluation of Anion Effect on the Solubility of Hydrogen Sulfide in Ionic Liquids Using Molecular Dynamics Simulation; etc.).

2. Please unify the unit of temperature in the text with K. The subscripts of H2S must be consistent throughout the text. 

3. The citation format is inconsistent, for example, the fourth reference is different from the others.

4. In Fig. 2, how did you control the pressure at different temperatures? The advances of the absorbent should be highlighted.

5. In Section 4, please add the references for the results obtained such as Fig. 7.

6. In Figs. 4 and 5, please detail the reasons for the turning points.

7. The full term of the abbreviations should be given such as LO-CAT.

8. This work only presented the pure gas experiments and modeling. How is the performance for a mixed gas containing H2S? The energy consumption should be considered for the comparisons of various absorbents such as MEA, amino acid salts, K2CO3, etc.

9. Please also improve the English writing in the current paper.

Round 2

Reviewer 1 Report

  • In lines 91-94 replace “99% (mass fraction) purity” by 99% mass purity or “of 97% (mass fraction) purity” by, of 97% mass purity.

  • The end of the conclusions must be improved “…in rFeCl3/[bmim]Cl. is the molar ratio of anhydrous FeCl3 to [bmim]Cl.”, it is confused.

  • The reference [10] is not complete, information are missing.

Author Response

In lines 91-94 replace “99% (mass fraction) purity” by 99% mass purity or “of 97% (mass fraction) purity” by, of 97% mass purity.

we have corrected it.

The end of the conclusions must be improved “…in rFeCl3/[bmim]Cl. is the molar ratio of anhydrous FeCl3 to [bmim]Cl.”, it is confused.

This is a clerical error. We have deleted the “is the molar ratio of anhydrous FeCl3 to [bmim]Cl.”.

The reference [10] is not complete, information are missing.

We have added the missing information.

Reviewer 3 Report

It is ok to publish this work.

Author Response

Thank you for your comments.

We have also improved the English in the latest submitted version.